# Comparative Investigation into the Roles of Imipenem:Cyclodextrin Complexation and Antibiotic Combination in Combatting Antimicrobial Resistance in Gram-Negative Bacteria

**DOI:** 10.3390/ph16101508

**Published:** 2023-10-23

**Authors:** Sara Mahmoud Farhan, Rehab Mahmoud Abd El-Baky, Hala Rady Ahmed, Zeinab Fathalla, Ali Alamri, Hamdy Abdelkader, Adel Al Fatease

**Affiliations:** 1Department of Microbiology and Immunology, Faculty of Pharmacy, Deraya University, Minia 11566, Egypt; sara.mahmoud@deraya.edu.eg (S.M.F.); rehab.mahmoud@mu.edu.eg (R.M.A.E.-B.); 2Department of Microbiology and Immunology, Faculty of Pharmacy, Minia University, Minia 61519, Egypt; halaradyahmed@yahoo.com; 3Department of Pharmaceutics, Faculty of Pharmacy, Minia University, Minia 61519, Egypt; zianab.mphamed@mu.edu.eg; 4Department of Pharmaceutics, College of Pharmacy, King Khalid University, Abha 62223, Saudi Arabia; aamri@kku.edu.sa (A.A.); habdelkader@kku.edu.sa (H.A.)

**Keywords:** proteus, klebsiella, *Acinetobacter baumannii*, drug combination, imipenem, amikacin, inclusion complex

## Abstract

Extensively drug-resistant (XDR), multidrug-resistant (MDR) and pandrug-resistant (PDR) Gram-negative microorganisms (GNBs) are considered a significant global threat. β-lactam and aminoglycoside combinations and imipenem:cyclodextrin inclusion complexes were studied for the treatment of lethal GNBs. This is because of the broad empiric coverage of the two drugs and their possession of different spectra of activity. Two cyclodextrins (β- and hydroxy propyl β-cyclodextrins) were utilized for inclusion complex formation with imipenem using the physical and kneading methods. In silico investigation using the molecular docking and Fourier-infrared spectroscopy (FTIR) were employed to estimate binding constant and confirm complex formation, respectively. The in vitro effects of amikacin and imipenem combination in comparison to the effect of imipenem-β- and hydroxy propyl β-cyclodextrin (CD) complexes against *Klebsiella* spp. and *Acinetobacter baumannii* were studied. The isolated microorganisms’ antimicrobial responsiveness to various antibiotics (19 antibiotics) was evaluated. It was found that piperacillin/tazobactam and gentamycin (resistance rates were 33.3% and 34%, respectively) were the most effective antimicrobials. The in vitro studies have been performed by the checkerboard technique and time-killing assay. The studied combination of amikacin and imipenem showed a substantial drop in bacterial count (*p* < 0.05). The in vitro studies demonstrated a synergism for the investigated combination. Conventional PCR was used in molecular studies to identify the resistance genes bla IMP and aac (6′)-Ib. The blaIMP and aac (6′)-Ib were recorded in 38.2% and 3.6% of the studied isolates, respectively. The in vitro studies showed synergistic effects among the tested antibiotics with FICIs of ≤0.5. Finally, the study compared the reduction in bacterial count between the tested antibiotic combinations and imipenem:CD physical and kneaded mixtures. Imipenem:CD inclusion complexes demonstrated a significant bacterial count reduction over the antibiotic combination. These results highlight the emerging role of CDs as safe biofunctional excipients in the combat against superbug bacterial resistance.

## 1. Introduction

Antimicrobial resistance is considered the most genuine problem worldwide. This problem can spread between countries and even continents; therefore, it is classified as a global problem rather than a local issue [1]. The abuse of antibiotic use in both human and animal, as well as growing industrialization, are the main causes of this global health threat [2]. Increasing Gram-negative antimicrobial resistance poses a significant threat to the treatment of hospital-acquired infections [3], leading to prolonged hospitalization, poor patient outcomes, and increased cost of treatment resulting in the increase in mortality rates [4].

Extensively drug-resistant (XDR) and pandrug-resistant (PDR) Gram-negative pathogens are viewed as a serious concern, represented mainly by *Pseudomonas aeruginosa, E. coli, Acinetobacter baumannii* and *Klebsiella pneumoniae* [5,6]. Extensively drug-resistant (XDR) bacteria are defined according to the ECDC and CDC as bacterial isolates that are resistant to one or more antibiotic drug classes; however, PDR is described as pathogenic bacteria resistant to all antibiotic categories [5].

Carbapenems still remain the only viable treatment option for severe infection [7,8]. Carbapenems are broad spectrum antibiotics; they prevent Gram-positive and Gram-negative resistant bacteria from synthesizing cell walls resulting in cell lysis and bacterial killing [9]. The β-lactam ring fused with a five-membered ring is essential for antibacterial activities; however, this bicyclic fused ring is one of the main reasons for the instability of carbapenems [10]. There are several members belonging to carbapenems that include imipenem, panipenem, meropenem, ertapenem, doripenem and biapenem; imipenem was selected for this study [10].

Due to their affinity for penicillin-binding proteins (PBPS), several beta lactam antibiotics, such as imipenem, may cause alterations in cell morphology, while aminoglycosides such as amikacin are protein synthesis inhibitors [11]. In addition, it was found that aminoglycosides affect the outer membrane structure, so it can facilitate the penetration of other antibiotics. Depending on the previous findings, synergistic combinations of these antibiotics can treat infections caused by XDR Enterobacteriaceae, *P. aeruginosa* or *A. baumannii* [12]. Carbapenem and amikacin combination was studied by many researchers. Yadav, Bulitta [13] reported the effect of meropenem on the destruction of the cell wall peptidoglycans that facilitate the entry of amikacin to inhibit protein synthesis. They found that meropenem and amikacin combination was more effective than colistin and amikacin combination against pseudomonas aeruginosa.

Many studies showed variable results concerning the sensitivity of XDR Gram-negative bacteria to combination antibiotics therapy [14]. The main goal of using antibiotic combinations is to reduce mortality and the rate of developing resistance. This can be achieved through different mechanisms which include bacterial clearance, suppression of bacterial resistance and synergism. On the flip side, some disadvantages of antibiotic combination with aminoglycosides have been reported such as increased systemic toxicity and greater cost [15]. In addition, they have a low bioavailability due to their hydrophilic properties and their tendency to be degraded in the gastric environment. A combination of β-lactam antibiotics with one of the fluoroquinolones or aminoglycoside antibiotics was utilized for the treatment of life-threatening infections such as sepsis [16]. Therefore, there is an imperative need to seek a safer and sustainable alternative approach for combating bacterial resistance.

It was reported that using biopolymers such as cyclodextrins in combination with carbapenems can increase stability and increase the antibacterial activity through improving the permeability of carbapenems [17]. Cyclodextrins are a natural group of cyclic oligosaccharides (six to eight sugar moieties) with hydrophobic cores and hydrophilic exteriors composing α-, β- and γ-cyclodextrins, respectively. Cyclodextrins can be schematically represented by a doughnut-like structure; the outside is hydrophilic and is strongly hydrated in aqueous solution. The inside is hydrophobic, therefore, water molecules located in the ring are repelled by the non-polar wall, due to hydrophobic interiors; this enables the cyclodextrins to extract/solubilize a range of guest molecules from the bulk aqueous solution that are water insoluble and have the right size and hydrophobicity [18]. More recently, cyclodextrins have been reported to enhance antibacterial activities and combat multidrug resistance [19,20]. Cyclodextrins can offer promising strategies against bacterial resistance. These include decreasing cell-to-cell communication, increasing permeability of antibacterial drugs through bacterial cell walls and depletion of cholesterol [21,22]. In addition, cyclodextrins have demonstrated inhibitory effects against efflux proteins by decreasing the activity of P-glycoproteins in the gastrointestinal membranes [23,24].

Hereby, we undergo a comparative study between the effects of imipenem and amikacin combination and imipenem with cyclodextrin complexes (β-cyclodextrin and hydroxypropyl β-cyclodextrin) on two of the most important pathogens (*Acinetobacter baumannii* and *Klebsiella pneumoniae*) isolated from patients with various infections in the upper Egypt region.

## 2. Results

Both physical mixing and kneading methods were adopted to prepare imipenem:cyclodextrins (CDs) complexes. These two methods are frequently utilized to create inclusion complexes of active pharmaceutical ingredients (APIs) and cyclodextrins [25]. The two methods do not involve heat or extreme freezing; nevertheless, the complexes can be formed at ambient conditions [26,27].

### 2.1. In Silico (Molecular Docking) Studies

In silico studies were employed to estimate the energy score and visualization of the inclusion complexes of imipenem with β-CD and HP β-CD. The visualization of the inclusion complexes, energy scores, type and number of the possible bonds’ formation are shown in Figure 1 and Table 1. The energy scores and cavity size could confirm formation of imipenem:CDs; nevertheless; formation of imipenem:HP β-CD was more favorable than imipenem:β-CD inclusion complexes. This was evident from the number of possible bonds and energy scores estimated for the two inclusion complexes. The energy score for imipenem:HP β-CD was greater (−6.357 Kcal/mol) than that (−5.908 Kcal/mol) for than imipenem:β-CD.

### 2.2. FTIR Spectroscopy

The FTIR spectra for the studied inclusion complexes prepared by physical and kneading techniques are shown in Figure 2 and Figure 3. The FTIR characteristic bands for imipenem were recorded; broad and medium intensity bands at 3500 cm^−1^ and 3200 cm^−1^ were attributed for alcohol (-OH) and secondary amine (-NH) stretching. The FTIR absorption band at 1640 cm^−1^ was due to imine (C=N) stretching and the acid (C=O) stretching appeared at 1688 cm^−1^ [28,29]. Broad peaks appeared at 3357 and 2923 cm^–1^ were due to O-H and C-H stretching of the cyclic sugar moieties of both β-CD and HP β-CD. The characteristic peaks of imipenem were either shifted or broadened due to H-bond formation and electrostatic interactions between carboxylic groups and the rims of CDs for Both FTIR spectra of imipenem:β-CD PM and imipenem:β-CD K indicating inclusion complexation interactions [30]. These results were in line with those obtained from the in silico (docking) studies in Section 2.1.

### 2.3. Bacterial Isolation and Identification

In this study, 150 Gram negative bacteria (GNB) were isolated from 200 samples collected from the below-mentioned infections. The majority of isolates were GNB and found in infections of the skin (62%), followed by chest infections (14.7%), urinary tract infections (8.7%), gastroenteritis (8.0%) and ear infections (6.6%). Ten isolates (6.67%) were positive for *Klebsiella* spp. and five isolates (3.33%) were positive for *A. baumannii* and other Gram-negative strains represented 90% of total isolates (Table 2).

### 2.4. Antibiotic Susceptibility Testing

The distribution of antibiotics among the resistant isolated microorganisms (Figure 4) showed that the highest resistance was shown against ampicillin/sulbactam, amoxicillin/clavulanic acid and cefotaxime.

### 2.5. Determination of MIC for Amikacin and Imipenem for Klebsiella spp. and Acinetobacter Baumannii Isolates

*Klebsiella* spp. and *Acinetobacter baumannii* showed the highest resistance values to amikacin (20%) as shown in Table 3. The distribution of the MIC values of imipenem among the isolated microorganisms showed that *Acinetobacter baumannii* was the most resistant microorganism to imipenem (40%), followed by *Klebsiella* spp. (30%), as shown in Table 4.

### 2.6. Molecular Assessment of aac(6’)-Ib and bla IMP by PCR

Fifteen different isolates from skin and chest infections were assessed, ten (66.7%) isolates were positive for *bla_IMP_* and one (6.67%) isolate was positive for *aac(6′)-Ib.* In *Klebsiella* species, six isolates harbored *bla_IMP_* from the total number of *Klebsiella.* The *bla_IMP_* was found in skin infections (4/6) and chest infections (2/6). For *A. baumannii,* four isolates harbored *bla_IMP_*. This gene was found only in the chest infections (2/4) and skin infections (2/4)_._ One isolate of *Klebsiella* species isolated from the chest infections harbored *aac(6′)-Ib* from the total number of *Klebsiella* (1/10) and for *A. baumannii*. All isolates were negative for *aac(6′)-Ib,*
Table 5.

### 2.7. Detection of the Synergism for Amikacin and Imipenem Combination against Klebsiella pneumoniae and Acinetobacter baumannii Resistant Isolates Using the Checkerboard Technique

The results indicated that amikacin and imipenem combination could be synergistically lower than the MICs of each drug alone. The FIC_index_ of both drugs was 0.023, which means that the combined drugs showed a synergistic activity. The combination against the tested resistant strains showed a high synergistic activity, with a high reduction in MIC for amikacin from 64 µg/mL to 0.5 µg/mL and for imipenem from 64 µg/mL to 1µg/mL in *Acinetobacter,* while by testing *Klebsiella,* a high reduction in MIC for amikacin from 512 µg/mL to 4 µg/mL and for imipenem from 64 to 1µg/mL was observed, Table 6.

### 2.8. Time–Kill Studies

Figure 5 shows the time killing of *Acinetobacter baumannii* profiles of imipenem alone (A), amikacin alone (B) and their combination (C) using four different MIC levels (0, 0.25, 0.5 and 1 MIC) over 24 h. At 0.25xMIC, the bacterial count decreased to 3.47 log_10_ CFU/mL, which meant a synergistic effect between both drugs. At 0.5xMIC, the count reduced to 2.63 log_10_ CFU/mL meaning bacteriostatic and synergistic activity. At 1xMIC the combination showed bactericidal action with a decrease in count to 5.6 log_10_ CFU/mL with 3 log_10_ CFU/mL reductions at 24 h (Figure 5 and Figure 6). Regarding the resistant *Klebsiella* sp., at 0.25xMIC the count decreased to 3 log_10_ CFU/mL; such a combination showed a synergistic activity between both drugs. At 0.50xMIC, 3.35 log_10_ CFU/mL reductions were recorded, indicating a bactericidal and synergistic activity for the combination of the two antibiotics at 24 h, while at the 1xMIC combination, a decrease in count after 8 h was observed with 3.6 log_10_ CFU/mL reductions (Figure 7 and Figure 8).

When it came to imipenem:CD complexes that prepared by the two methods: physical mixing (PM) and kneading (K) techniques; the results interestingly showed superior antibacterial activity for imipenem in complex with the two investigated CDs (β-CD and HP β-CD) for both bacterial isolates. The findings recorded more significant decreases in colony counts for imipenem:CDs complexes than that shown by the imipenem–amikacin combination. There were no statistically significant (*p* > 0.05) differences between imipenem:β-CD and imipenem:HP β-CD complexes.

Regarding *Acinetobacter baumannii,* the colony counts for imipenem alone at 0.25xMIC was reduced by 3.8 log_10_ CFU/mL; at 0.5xMIC colonies were reduced by 3.96 log_10_CFU/mL and at 1xMIC, the reduction was 4.44 log_10_ CFU/mL. For imipenem:β-CD PM at 0.25xMIC, the count was reduced by 4 log_10_ CFU/mL; while at 0.5xMIC, the colonies were reduced by 4.96 log10CFU/mL. The greatest reduction was recorded at 1xMIC by 5.12 log_10_ CFU/mL. In imipenem:β-CD K at 0.25xMIC bacterial count recorded a significant (*p* < 0.05) decrease by 4.02 log_10_CFU/mL, compared to imipenem alone; while at 0.5xMIC the colonies were reduced by 4.06 log_10_CFU/mL. At 1xMIC, the reduction was 4.32 log_10_ CFU/mL. In imipenem:HP β-CD K at 0.25xMIC recorded lower colonies by 4.16 log_10_CFU/mL; while at 0.5xMIC, the colonies were reduced by 4.35 log_10_CFU/m. At 1xMIC, the reduction was 5.06 log_10_ CFU/mL.

For imipenem:HP βCD PM at 0.25xMIC, the count was reduced by 4.16 log_10_ CFU/mL; while at 0.5xMIC, the colonies were reduced by 4.31 log_10_ CFU/mL. The highest reduction was recorded at 1xMIC by 5.34 log_10_ CFU/mL (Figure 9, Figure 10 and Figure 11).

For *Klebsiella*, imipenem alone at 0.25xMIC colonies were decreased by 1.84 log_10_ CFU/mL; while at 0.5xMIC colonies were reduced by 0.34 log_10_ CFU/mL and at 1xMIC the reduction was 2.46 log_10_CFU/mL. For imipenem:β-CD PM at 0.25xMIC bacterial count was reduced by 3.72 log_10_ CFU/mL; while at 0.5xMIC, the colonies were decreased by 4.96 log_10_ CFU/mL. At 1xMIC the reduction was 4.96 log_10_ CFU/mL. In imipenem:β-CD PM K at 0.25xMIC the count was decreased by 3.35 log_10_ CFU/mL; while at 0.5xMIC the colonies were reduced by 4.97 log_10_ CFU/mL. At 1xMIC, the reduction was 5.14 log_10_ CFU/mL.

For imipenem:HP β-CD K at 0.25xMIC, the count decreased by 3.27 log_10_ CFU/mL; while at 0.5xMIC colonies were reduced by 4.61 log_10_ CFU/mL after 24 h and at 1xMIC the reduction was 6.36 log_10_ CFU/mL after 24 h. In PM at 0.25xMIC colonies were reduced by 3.97 log_10_ CFU/mL after, at 0.5xMIC colonies were decreased by 4.96 log_10_ CFU/mL and at 1xMIC the reduction was 6.11 log_10_ CFU/mL (Figure 12, Figure 13 and Figure 14).

It was found that the antibiotic combination reduced the *Acinetobacter baumannii* viable colony counts by 2.63 to 3 log_10_ CFU/mL, while imipenem:CD inclusion complexes recorded reduction ranged in the count from 3.8 to 5.12 log_10_ CFU/mL. For *Klebsiella* isolate, antibiotic combination reduced the viable count by 3 to 3.6 log_10_ CFU/mL while imipenem:CD inclusion complexes reduced the viable count by 3.35 to 6.11 log_10_ CFU/mL. These results showed that the tested imipenem:CD inclusion complexes showed better synergistic activity than the tested combinations and each antibiotic alone.

## 3. Discussion

Antimicrobial resistance is considered a worldwide problem as it represents a great challenge to the public health. The most important risk factor for extensive resistance in GNBs is the exposure to antimicrobial agents for the long term [31]. Antimicrobial resistance has become a major problem of great global concern. In some third-world countries, patients can obtain antimicrobials without prescription and ineffective dose regimens can lead to the selection of resistant cells, resulting in therapy failure; these malpractices can cross borders through traveling individuals. In addition, using bactericidal antibiotics for livestock to protect animals from infection could contribute to the bacterial resistance and dissemination of antimicrobial resistance and resistance genes among strains in the same environment [32]. In addition, the prevalence of resistance genes of β-lactams is mainly due to the higher utilization rates of β-lactams of different classes in hospitals and community pharmacy. All these factors differ from one country to another, according to the applied guidelines in using antibiotics.

The emergence of this resistance can be overcome or delayed using combination therapy. Aminoglycosides such as amikacin can suppress protein synthesis, while other beta lactam antibiotics, such as imipenem, exhibit a certain affinity for penicillin-binding proteins (PBPSs) and cause alterations in cell shape. However, combinations with aminoglycosides have been associated with ototoxicity and renal adverse events [33]. This study was carried out to assess the combination therapy for imipenem and amikacin in the treatment of XDR Gram-negative pathogens in comparison to imipenem:CD complexes with β-CD and HP β-CD prepared using two methods physical mixing and kneading methods. In contrast to potential toxicity could occur due to using aminoglycosides, cyclodextrins are natural excipients that have been used extensively to solve biopharmaceutics problems such as drug solubility and permeability. In this study, the capacity of CDs to improve antibiotics efficacy and combat bacterial resistance in XDR, MDR and PDR Gram-negative microorganisms was studied. Imipenem:CD inclusion complexes were prepared using two different methods to test their activity on the tested strains. Two CDs, the native β-CD and the semisynthetic derivative HP β-CD, were studied. There are cumulative reports highlight the high safety profiles, the reported ability of CDs to increase antibiotics stability and increasing carbapenem antibiotic permeability across bacterial membranes [16,17]. Molecular docking recorded binding constants of −5.908 and −6.357 for imipenem:β-CD and imipenem:HP β-CD, respectively. These binding constants can ensure inclusion complexation and have been confirmed elsewhere using thermal and spectral analyses [26,34].

Wareham, Momin [35] found that the most predominant GNB was *Acinetobacter baumannii*, while Aygun, Aygun [36] found that *Klebsiella* was the most common isolate. The current study revealed that most resistance activity was shown against ampicillin/sulbactam, amoxicillin/clavulanic acid and cefotaxime. On the contrary, the opposite results were reported by Vena, Giacobbe [37], Kofteridis, Andrianaki [38], and Sharahi, Ahovan [39].

The *Bla_IMP_* and *aac(6′)-Ib* genes are considered the most abundant β-lactams and aminoglycosides genes in GNBs [40]. Therefore, the investigated isolates were assessed for their presence. A study discussed by Elbadawi, Elhag [41] and Manohar, Leptihn [42] showed that 2% of Gram-negative isolates harbored *bla_IMP_*. Another study carried out by Gajamer, Bhattacharjee [43] found that *bla_IMP_* was found only in 1.1% isolates of the total Gram-negative bacteria. The difference in the incidence of resistance genes among different studies depends on the uncontrollable use of antibiotics in livestock and human and horizontal gene transfer among strains in different environments.

A study conducted by Firmo, Beltrão [44] reported that out of the 35 resistant Gram-negative isolates, 34 (97.1%) harbored aminoglycoside modifying genes and most prevalent AME genes were *aac(6’)-Ib* (42.9%), *aac(3)-IIa* (40.0%) and *ant(2”)-Ia* (20.0%). These results were in agreement with our results as aac(6’)-Ib was detected in the same percentage. Another study performed by Costello, Deshpande [45] found that among Gram-negative clinical isolates, the *aac(6’)-lb* gene was the most predominant AME gene and were found in a total of 72 isolates (36.0%).

Carbapenems can be used as monotherapy in hospital acquired infections and can be used in combination with other antibiotics such as aminoglycosides. Many studies discussed the effect of combination therapy of carbapenems and aminoglycosides. Aminoglycosides have a marked antimicrobial activity against GNBs as monotherapy. However, the rapid emergence of resistance strains and nephrotoxicity reported with aminoglycosides could offset their benefits for clinical use.

It was reported that aminoglycosides showed disruptive effects on the outer membrane structure by binding to the negatively charged lipopolysaccharides in the outer membrane of GNBs. Therefore, it can facilitate the permeability of other antibiotics such as carbapenems when used in combination. Depending on the previous findings, antibiotic combination therapy can decrease the chance of developing resistance upon using them as monotherapies [46,47,48]. The in vitro activity of amikacin and imipenem each alone and in combination against selected *bla_IMP_* and *aac(6’)-Ib* producing GNBs was evaluated in the present study. The main motive to study these medications was made since there were scarce published clinical trials evaluating the effectiveness of combining amikacin with imipenem, on clinically relevant GNB isolates from chest and skin infections.

Variable results were shown according to the type of the tested organisms and the represented resistance mechanisms [46,49,50]. Uddin, Saha [51] found that the combination of imipenem with amikacin showed 54.2% synergistic activities, 20.8% additive activities, but exhibited insignificant effects against 25% of imipenem resistant *A. baumannii*. In addition, no antagonism activity was detected. Shabayek, El-Damasy [52] demonstrated an additive effect between meropenem and amikacin in the treatment of multi-drug resistant *Acinetobacter*. The combination efficacy of meropenem and amikacin against metallo-β-lactamases producing *Acinetobacter* strains was discussed by Esadoglu, Ozer [53]. They found an additive effect against 51% of strains, a synergistic effect against 49% of the tested isolates while no antagonistic interaction was found with the tested combination These results may be due to the presence of different methods of resistance against the same antibiotic by different microorganisms.

In our investigation, an amikacin and imipenem combination had the ability to lower the MICs less than the susceptibility breakpoint in all tested GNBs. The study illustrated that the combination reduced the final bacterial counts more effectively than that shown by each treatment alone, lessened the risk of monotherapy developing resistance and decreased the stress of clinical treatment, making it a possible therapeutic alternative for the treatment of lethal infections caused by Gram-negative pathogens.

Due to the neurotoxic and the ototoxic side effects of aminoglycosides, most drug combination trials have failed to exhibit sufficient safety and efficacy [54]. The design and conduct of early-phase combination trials present specific challenges, such as determining which agents to combine, choosing an appropriate dose and schedule (including which agent to escalate), and addressing drug–drug interactions and overlapping toxicities [55]. So, the recent study compared the effect of an imipenem and amikacin combination with imipenem:CD physical mixtures (PMs). The imipenem:CD PMs revealed a more significant (*p* < 0.05) decrease in colony count than imipenem and amikacin combination. In addition, this study discussed that the use of these mixtures in the future may be better than the combination therapy of two antibiotics.

A study carried out by Paczkowska and his colleague found that mixture of carbapenem such as tebipenem with cyclodextrins decreases the MIC in Gram-positive and Gram-negative bacteria due to the ability of CDs to block porin channels contributing to the efflux effect in bacteria and its ability to increase membrane permeability to antibiotics [56]. A recent study revealed that cyclodextrin improves physicochemical and pharmacological properties with better efficacy when used in combination with antimicrobials as new alternatives [57,58]. When used as complexing agents, CDs can also increase an antibiotic’s solubility and enhance drug permeability through the membrane barrier, thus improving the bioavailability of the guest molecule, and modifying the antibacterial activity and chemical stability [56]. Irrespective of the preparation methods, the results indicated that both β-CD and HPβ-CD equally enhanced antimicrobial activities of imipenem against Gram-negative isolates [59]. It is worth mentioning that there were no statistically significant differences recorded for the antibacterial enhancing effects from the two studied imipenem:β-CD and imipenem:β-CD complexes, as indicated by the time killing profiles for the two studied GNBs. These findings could indicate that both β-CD and HPβ-CD were equally effective for improving imipenem efficacy toward the resistant bacteria.

## 4. Materials and Methods

Imipenem monohydrate (purity 95%) was purchased from Pharma Quanao Chemical Co., Ltd., Guangdong, China. Beta (β)-cyclodextrin, and hydroxypropyl β-cyclodextrin were supplied by Acros Organics, NJ, USA.

### 4.1. Preparation of Imipenem:Cyclodextrin Physical Mixtures and Kneaded Dispersions

Equivalent amounts (mg) of molar weight of imipenem (299.35 mg), β-CD (1135 mg) and HP β-CD (1400 mg) physical mixtures were weighed separately and mixed thoroughly in a watch glass for 5 min using a spatula and passed through a 125-µm sieve to prepare physical mixtures. The mixtures were stored in airtight glass containers and were kept in a cool place until further use.

The kneading dispersions were prepared as mentioned above and then a hydro-methanolic solution (50% *v*/*v*) was dropped (approximately 0.5 mL) on the mixtures and kneaded till a cohesive dough was obtained. The kneaded mixtures were allowed to dry at ambient conditions for 4 h, then pulverized and sieved through a 125 µm sieve and stored in a cool place until further use.

### 4.2. Molecular Docking

Molecular Operating Environment (MOE) 2014.09 software (Chemical Computing Group, Montreal, QC, Canada) was used to investigate the docking, energy scores and potential orientation of imipenem within the cavity and/or rim of β-CD and HP β-CD. The 3D structure of β-CD was downloaded from Protein Data Bank (PDB) at https://www.rcsb.org (1 June 2022) as PDB file code: 5E6Z [21]. The 3D structure of HP β-CD was designed using the builder interface of MOE software via substituting the hydroxyl group with isopropyl radicals [22]. Compounds were docked into the cavity of the CDs using an induced-fit docking protocol using the Triangle Matcher method and dG scoring system for pose ranking.

### 4.3. FTIR Spectroscopy

FTIR spectrophotometry (Spectrum Two, FTIR spectrophotometer, PerkinElmer Inc., Waltham, MA, USA) and Spectrum TM 10 software were used to stack the spectra for imipenem, β-CD, HP β-CD and their physical and kneaded mixtures. The spectra were collected directly from the dispersed powder on the diamond surface. The spectra were collected in a range of 4000 to 400 cm^−1^.

### 4.4. Bacterial Isolates

A total of 200 clinically relevant samples were gathered from patients suffering from different infections admitted to El-Minia University hospitals between August 2022 and February 2023. Different infections were covered in this study include skin infections, ear, chest, urinary tract infection (UTI), and gastrointestinal tract (GIT) infections. Gram-negative bacteria (GNB) isolation and identification were carried out employing conventional microbiological and biochemical assays [60]. Our study was performed on imipenem resistant isolates of *Klebsiella pneumoniae* and *Acinetobacter baumanii*. After isolation, the isolates were preserved in 20% glycerol at −20 °C.

### 4.5. Antimicrobial Sensitivity Test

The disc diffusion method [30] was employed for studying antibiotic sensitivity by using different antimicrobial categories (Oxoid, Basingstoke, UK) [31]. Imipenem (10 µg), Tobramycin (10 µg), aztreonam (30 µg), piperacillin (100 µg), piperacillin/tazobactam (10µg), gentamycin (10 µg), norfloxacin (10 µg), amoxicillin/clavulanic acid (30 µg), ampicillin/sulbactam (20 µg), ceftazidime (30 µg), cefotaxime (30 µg), ceftriaxone (30 µg), cefepime (30 µg), meropenem (10 µg), ciprofloxacin (5 µg), amikacin (30 µg) and levofloxacin (5 µg), ofloxacin (10 µg) and azithromycin (30 µg) were investigated. Further, minimum inhibitory concentrations (MICs), MIC90 and MIC50 for both imipenem and amikacin and imipenem:CD complexes were assessed against all isolates using the broth microdilution method [31].

### 4.6. Detection of aac(6′) -Ib and bla _IMP_ by Conventional PCR

A procedure described by Wilson [32] was used to extract DNA. The PCR was run in a thermal cycler (Bio Rad, Hercules, CA, USA). A 25 µL PCR mixture was used for the amplification including (0.2–0.400 µg) of the DNA sample, 12.5 µL of PCR master mix (0.5 M of Tris/HCl pH 8.55, 1.5 mM of MgCl_2_, 0.2 mM of dNTPs and 0.04 units/uL of Taq DNA polymerase), 1 µL of each forward and reverse primers and deionized water to a volume of 25 µL. PCR conditions are listed in Table 7.

### 4.7. Checkerboard Testing Method for Imipenem/Amikacin Combination

The checkerboard testing method was performed to identify potential synergistic effects of the antibiotic combination. This method depends on the microdilution of tested antibiotic combinations by using a dilution range from 64 µg/mL to 30 ng/mL using 10-fold dilution. The inoculation was obtained from bacterial colonies grown overnight on the Muller Hinton agar with an OD of 1, which is correlated to 1.5 × 10^6^ CFU/mL^−1^. The recorded in vitro activities among the investigated antibiotics were expressed by the fractional inhibitory concentration (FIC) that was estimated through the equation as below:FIC index = [FIC]_A_ + [FIC]_B_


FIC index of ≤0.5 indicated a synergism between drug A and drug B, while an FIC value greater than 4 indicated antagonism and FIC of > 0.5 ≤ 4 indicated additivity [49].

### 4.8. Time-Killing Assay for Impinem/Amikacin Combination and Imipenem:CD Mixtures

Time-killing plots were constructed to evaluate the bactericidal activity of imipenem/amikacin combination (0.25xMIC, 0.5xMIC and 1xMIC) and imipenem:CDs mixtures (0.25xMIC, 0.5xMIC and 1xMIC). Different concentrations were investigated in both separate drug and combination studies. The assay was performed by adding a suspension of the tested bacteria (0.5 MacFarland), which was then incubated for 1 day at 37 °C. Ten-fold dilutions were cultured on the sheep blood agar every 0, 2, 4, 8 and 24 h after incubation [64]. A bacteriostatic mechanism was indicated with a value of ≥2 log_10_; however, <3 log_10_ reductions in CFU/mL at 24 h compared to the initial inoculum and bactericide were found in the presence of ≥3 log_10_ reductions in CFU/mL at 24 h, in comparison to the starting inoculum, while a decrease of 2 log_10_ in CFU/mL was observed with the antibiotic combination, which is defined as a synergistic effect [65].

### 4.9. Statistical Analysis

The data were statistically analyzed employing the SPSS software, v. 25. Microsoft Office Excel 365 software was used for the graphical presentation. A Shapiro–Wilk test was performed for quantitative data between different groups or concentrations by applying one-Way ANOVA test followed by Post Hoc Tukey’s test. A repeated ANOVA test followed by Post Hoc LSD was applied with *p* < 0.05.

## 5. Conclusions

Cyclodextrins (e.g., β-CD and HP β-CD) are natural excipients that have been used extensively to solve biopharmaceutic problems such as poor drug solubility and permeability. In this study, potential new roles of CDs to enhance bacterial efficacy and combat bacterial resistance in XDR, MDR and PDR Gram-negative microorganisms were studied as potential alternatives to combination therapy, which are not devoid of toxicity and limitations. Irrespective of the preparation techniques used (physical and kneading methods), imipenem:CD mixtures showed a better reduction in colony counts than antibiotic combinations against the isolated Gram-negative bacteria from chest and skin infections. More interestingly, both imipenem:β-CD and imipenem:HPβ-CD complexes showed an equally enhanced antibacterial activity of the antibiotic drug toward the resistant strains. Our in vitro results demonstrated a great benefit upon using imipenem:CD mixtures over the tested combination in treatment of infections caused by these pathogens. This study recommended the consider these potential natural CD mixtures instead of antibiotics combination in the future.

## Figures and Tables

**Figure 1 pharmaceuticals-16-01508-f001:**
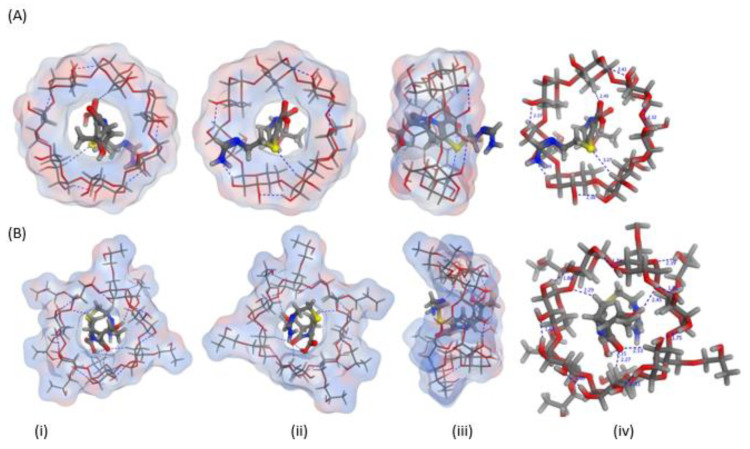
Three-dimensional orientations for imipenem docked into the cyclodextrin (CD) cavity of (**A**) β-CD and (**B**) hydroxypropyl (HP)-β-CD showing the (i) top, (ii) bottom (iii) side and (iv) top views with marked interactions showing the bond lengths.

**Figure 2 pharmaceuticals-16-01508-f002:**
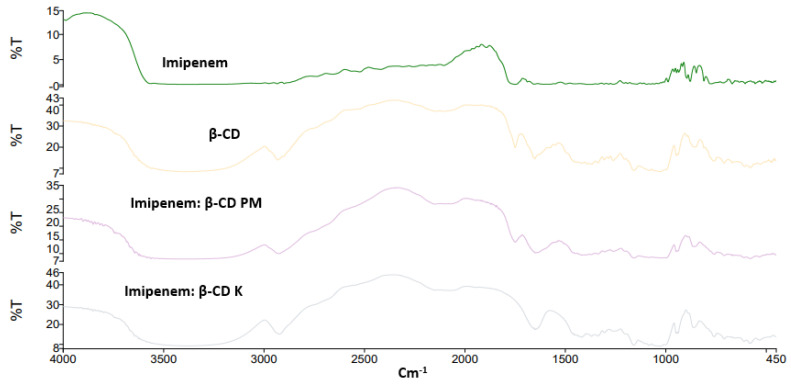
FTIR spectra for imipenem, β-cyclodextrin (CD), imipenem:β-CD physical mixture (PM) and imipenem:β-CD kneaded mixture (K).

**Figure 3 pharmaceuticals-16-01508-f003:**
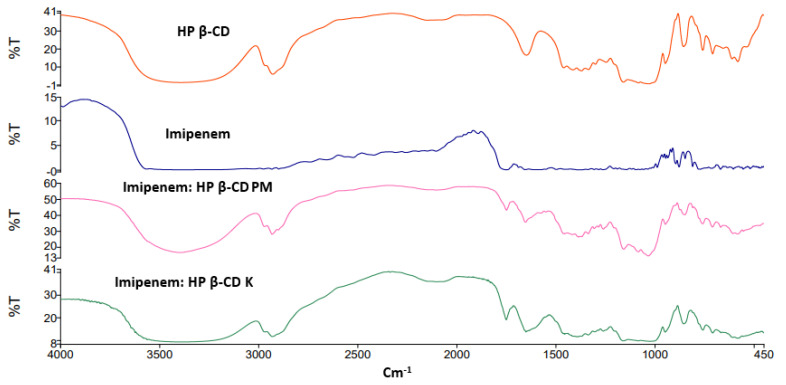
FTIR spectra for imipenem, hydroxy propyl (HP) β-cyclodextrin (CD), imipenem:HP β-CD physical mixture (PM) and imipenem:HP β-CD kneaded mixture (K).

**Figure 4 pharmaceuticals-16-01508-f004:**
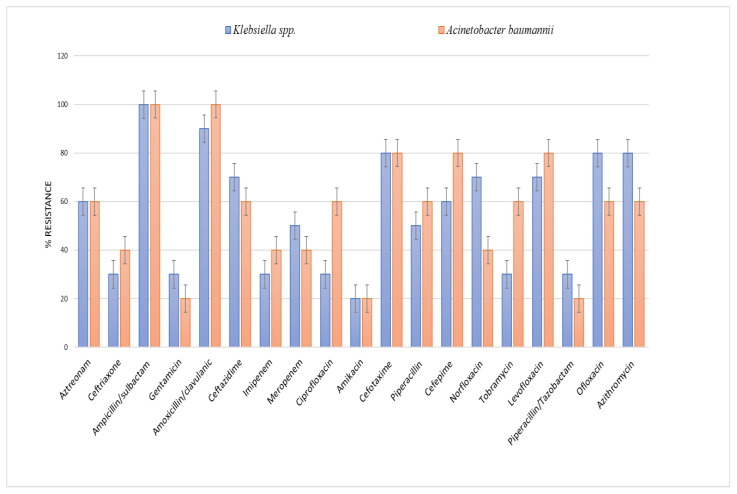
Antibiotic resistance of *Klebsiella* spp. *and Acinetobacter baumannii* isolates against different antibiotics.

**Figure 5 pharmaceuticals-16-01508-f005:**
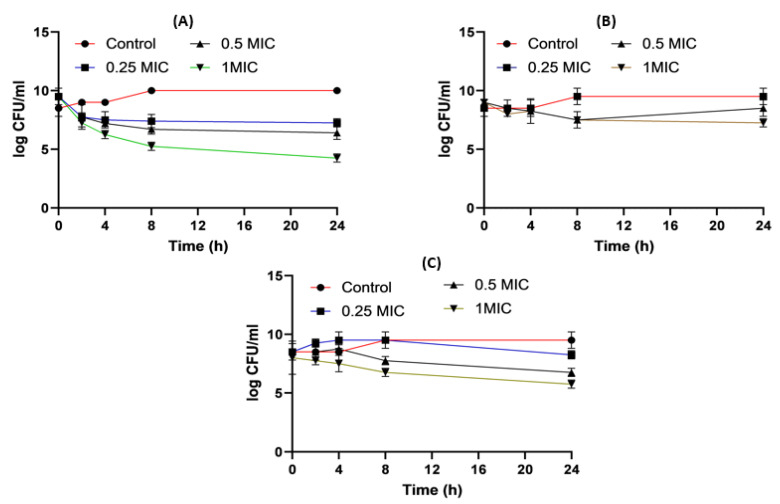
*Acinetobacter baumannii* time killing profiles for imipenem alone (**A**), amikacin alone (**B**) and imipenem and amikacin combination (**C**) at four different minimum inhibitory concentrations: 0, 0.25, 0.5 and 1 MIC (µg/mL). Data represent means ± SD.

**Figure 6 pharmaceuticals-16-01508-f006:**
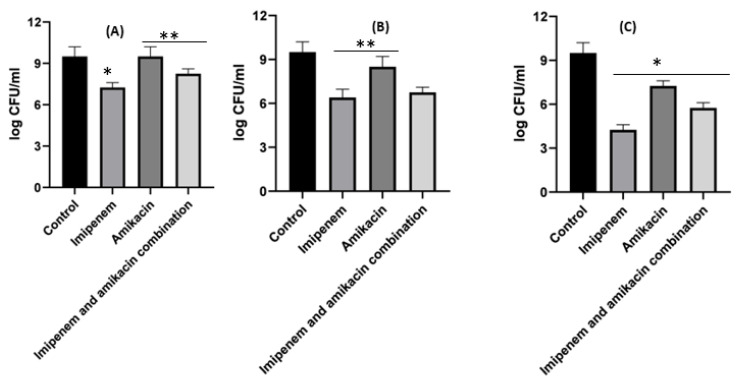
Different MIC (µg/mL) at 24 h for imipenem, amikacin and a combination of imipenem and amikacin at 0.25 MIC (µg/mL) (**A**), 0.5 MIC (µg/mL) (**B**) and 1 MIC (µg/mL) (**C**) using *Acinetobacter baumannii*. Data were extracted from Figure 5A–C for comparison purposes. * Denotes a statistically significant difference *p* < 0.05; ** denotes statistically non-significant difference *p* > 0.05.

**Figure 7 pharmaceuticals-16-01508-f007:**
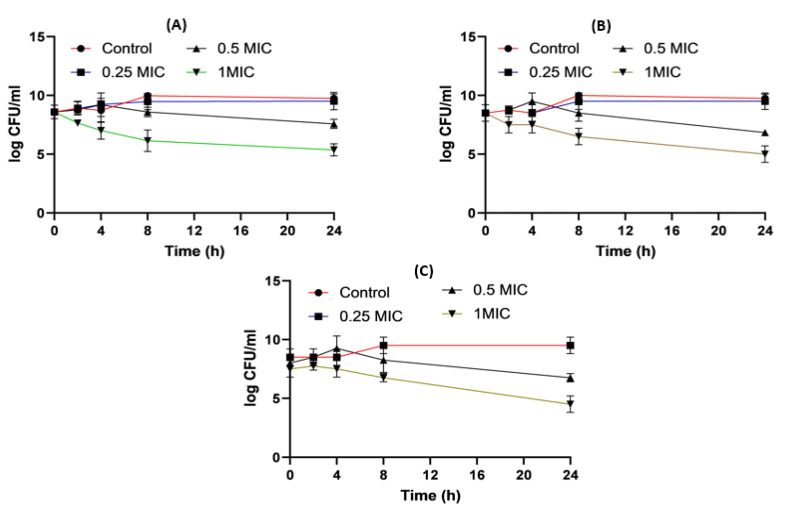
*Klebsiella* time killing profiles for imipenem alone (**A**), amikacin alone (**B**) and imipenem and amikacin combination (**C**) at four different minimum inhibitory concentrations: 0, 0.25, 0.5 and 1 MIC (µg/mL). Data represent means ± SD.

**Figure 8 pharmaceuticals-16-01508-f008:**
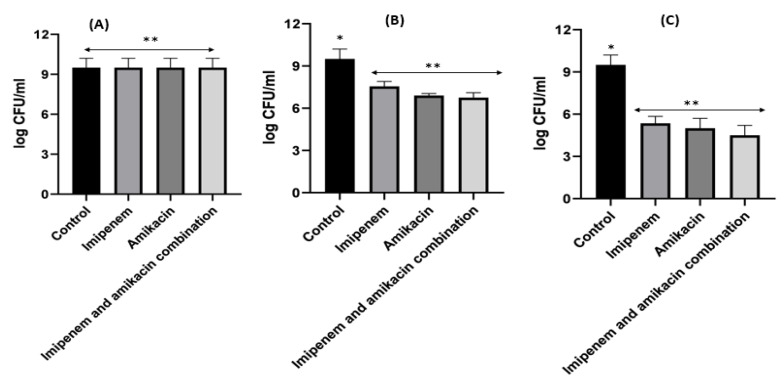
Different MIC (µg/mL) at 24 h for imipenem, amikacin and combination of imipenem and amikacin at 0.25 MIC (µg/mL) (**A**), 0.5 MIC (µg/mL) (**B**) and 1 MIC (µg/mL) (**C**) using *Klebsiella*. Data were extracted from Figure 7 A–C for comparison purposes. * Denotes statistically significant difference *p* < 0.05; ** denotes statistically non-significant difference *p* > 0.05.

**Figure 9 pharmaceuticals-16-01508-f009:**
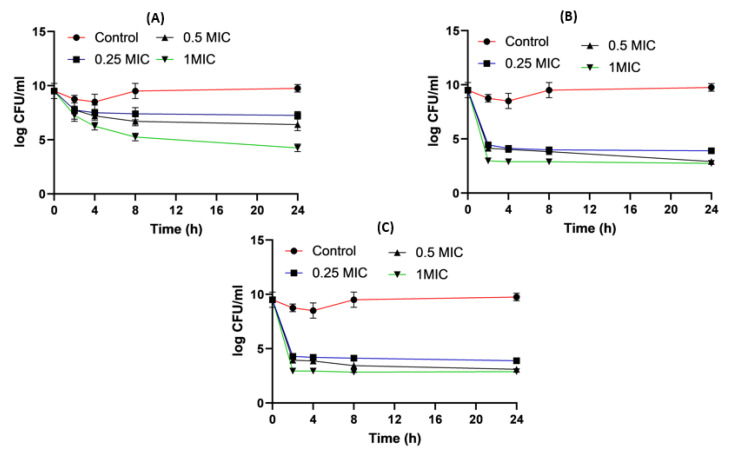
*Acinetobacter baumannii* time killing profiles for imipenem alone (**A**), imipenem:β-CD PM (**B**) and imipenem:β-CD K (**C**) at four different minimum inhibitory concentrations: 0, 0.25, 0.5 and 1 MIC (µg/mL). Data represent means ± SD.

**Figure 10 pharmaceuticals-16-01508-f010:**
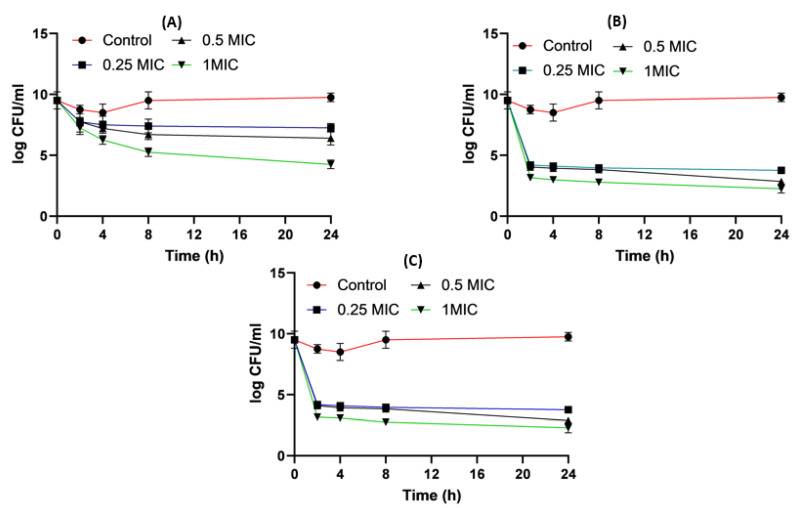
*Acinetobacter baumannii* time killing profiles for imipenem alone (**A**), imipenem:HP β-CD PM (**B**) and imipenem:HP β-CD K (**C**) at four different minimum inhibitory concentrations: 0, 0.25, 0.5 and 1 MIC (µg/mL). Data represent means ± SD.

**Figure 11 pharmaceuticals-16-01508-f011:**
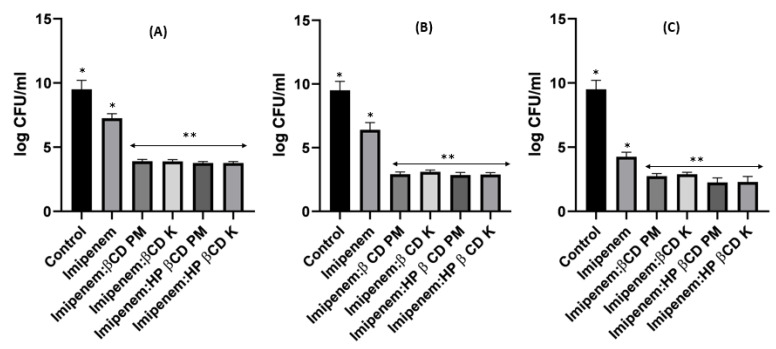
Illustrative histograms of imipenem, imipenem:β-CD PM, imipenem:β-CD K, imipenem:HP β-CD PM and imipenem:HP β-CD K at 0.25 MIC (µg/mL) (**A**), 0.5 MIC (µg/mL) (**B**) and 1 MIC (µg/mL) (**C**). Data were extracted from Figure 9 and Figure 10 for comparison purposes. * Denotes statistically significant difference *p* < 0.05; ** denotes statistically non-significant difference *p* > 0.05.

**Figure 12 pharmaceuticals-16-01508-f012:**
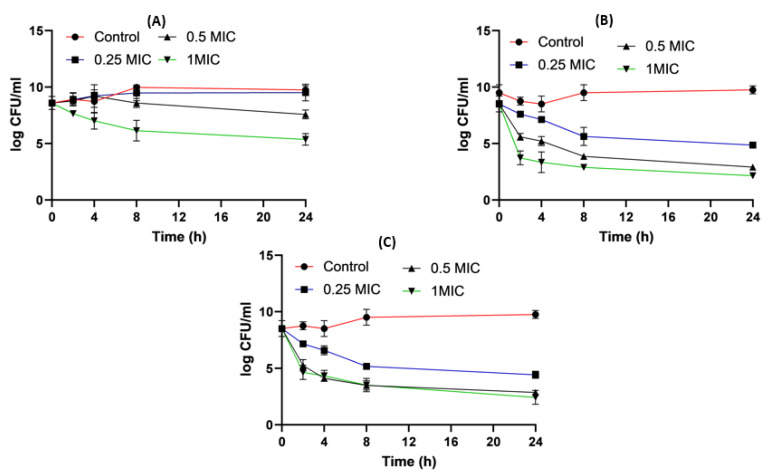
*Klebsiella* time killing profiles for imipenem alone (**A**), imipenem:β-CD PM (**B**) and imipenem:β-CD K (**C**) at four different minimum inhibitory concentrations: 0, 0.25, 0.5 and 1 MIC (µg/mL). Data represent means ± SD.

**Figure 13 pharmaceuticals-16-01508-f013:**
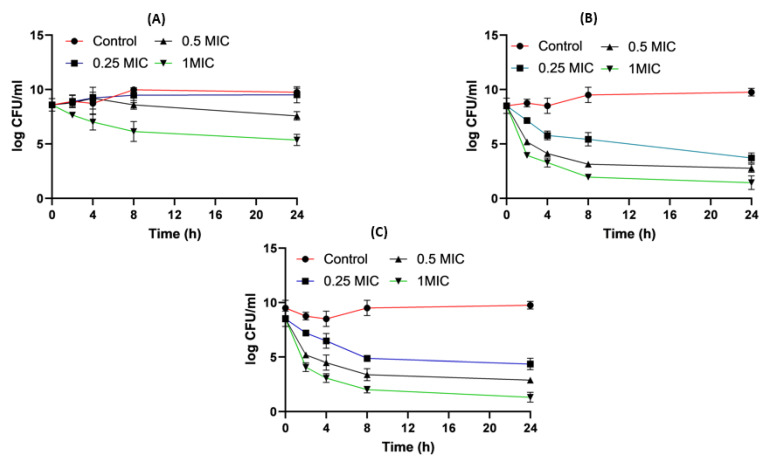
*Klebsiella* time killing profiles for imipenem alone (**A**), imipenem:HP β-CD PM (**B**) and imipenem:HP β-CD K (**C**) at four different minimum inhibitory concentrations: 0, 0.25, 0.5 and 1 MIC (µg/mL). Data represent means ± SD.

**Figure 14 pharmaceuticals-16-01508-f014:**
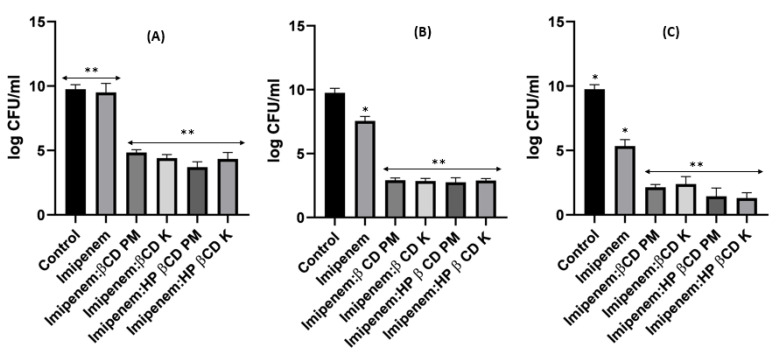
Illustrative histograms of imipenem, imipenem:β-CD PM, imipenem:β-CD K, imipenem:HP β-CD PM and imipenem:HP β-CD K at 0.25 MIC (µg/mL) (**A**), 0.5 MIC (µg/mL) (**B**) and 1 MIC (µg/mL) (**C**). Data were extracted from Figure 12 and Figure 13 for comparison purposes. * Denotes statistically significant difference *p* < 0.05; ** denotes statistically non-significant difference *p* > 0.05.

**Table 1 pharmaceuticals-16-01508-t001:** Energy score, type and number of possible interactions visualized for imipenem docked into the inclusion pocket of β-CD and HP β-CD.

API	Cyclodextrins	Energy Score (Kcal/mol)	Number and Types of Interactions	
			H-Bond	Hydrophobic Attractions
Imipenem	β-cyclodextrin	−5.908	2	-
	Hydroxypropyl-β-cyclodextrin	−6.357	4	-

**Table 2 pharmaceuticals-16-01508-t002:** The frequency of identified Gram-negative bacteria among various illnesses.

Type of the Infection	Isolate Counts	*Klebsiella* spp.	*A. baumannii*	Other Gram-Negative Bacteria
Skin infections	93	5	3	85
Ear infections	10	-	-	10
Chest infections	22	5	2	15
Urinary tract infections	13	-	-	13
Gastroenteritis	12	-	-	12
Total (%) *	150 (100)	10 (6.67)	5 (3.33)	135 (90%)

* Percentage (%) was estimated as the total isolate count.

**Table 3 pharmaceuticals-16-01508-t003:** Distribution of MIC, MIC_90_ and MIC_50_ of amikacin against the isolated *Klebsiella* spp. and *Acinetobacter baumanni*.

No.	MIC (µg/mL)													MIC_90_	MIC_50_	R	%
	0.25	0.5	1	2	4	8	16	32	64	128	256	512	1024				
*Klebsiella* spp. (n = 10)	0	0	6	2	2	0	0	0	0	0	1	1	0	4	1	2	20
*A. baumannii* (n = 5)	0	0	2	1	1	1	0	0	1	0	0	0	0	8	2	1	20

**Table 4 pharmaceuticals-16-01508-t004:** Distribution of MIC, MIC_90_ and MIC_50_ of imipenem among the isolated *Klebsiella* spp. and *Acinetobacter baumannii*.

No.	MIC (µg/mL)													MIC_90_	MIC_50_	R	%
	0.25	0.5	1	2	4	8	16	32	64	128	256	512	1024				
*Klebsiella* spp. (n = 10)	0	0	5	2	0	0	0	0	3	0	0	0	0	64	1	3	30
*A. baumannii* (n = 5)	0	0	2	1	0	0	0	0	2	0	0	0	0	64	2	2	40

**Table 5 pharmaceuticals-16-01508-t005:** Distribution of *bla_IMP_* genotype among the resistant isolates.

Name of Organism	Source of Sample	No. of Isolates in Each Infection	Resistance Pattern of Imipenem			*bla_IMP_*	*aac(6′)-Ib*
			Sensitive	Intermediate	Resistance		
*Klebsiella* spp.	Skin infections	5	2	1	2	4	-
	Chest infections	5	2	2	1	2	1
*Acinetobacter baumanii*	Skin infections	3	2	-	1	2	-
	Chest infections	2	-	1	1	2	-

**Table 6 pharmaceuticals-16-01508-t006:** MIC and FIC for imipenem alone, amikacin alone and their combination against selected *Klebsiella pneumoniae* and *Acinetobacter baumannii* resistant isolates.

	MIC (µg/mL)				FIC_amikacin_	FIC_imipenem_	FIC_index_	Outcome
	Amikacin Alone	Imipenem Alone	Amikacin + Imipenem Combination					
*Klebsiella isolate* (No. 9)	512	64	4	1	0.0078	0.0156	0.023	Synergistic
*A.baumannii* (No. 4)	64	64	0.5	1	0.0078	0.0156	0.023	Synergistic

**Table 7 pharmaceuticals-16-01508-t007:** The primers employed in the study.

Genes	Sequence	Temperature (°C)	Product Size	References
*bla_IMP_*	F:CATGGTTTGGTGGTTCTTGT	59	488	[61]
	R:ATAATTTGGCGGACTTTGGC			
*aac(6′)-Ib*	F:AGTACTTGCCA GCGTTTTAGCGC	51	365	[62]
	R:CATGTACACGGCTGGACCAT			
16S rRNA	F: GCTGACGAGTGGCGGACGGG	55	253	[63]
	R:TAGGAGTCTGGACCGTGTCT			

## Data Availability

Data is contained within the article.

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
