# Peer review of "Comparative Investigation into the Roles of Imipenem:Cyclodextrin Complexation and Antibiotic Combination in Combatting Antimicrobial Resistance in Gram-Negative Bacteria"

_pharmaceuticals, 2023, doi:10.3390/ph16101508_

Round 1
Reviewer 1 Report
Dear author,
The paper entitled “Comparative investigation into The Roles of Imipenem:Cyclodextrin Complexation and Antibiotic Combination in Combatting Antimicrobial Resistance in Gram-Negative Bacteria” has been intensively reviewed and evaluated. Although present study was considered an interesting study, there were some points that need to be revised. Hereby, I would like to present my suggestions and revisions.
Revision_1: before publication, the paper must be formatted according to the journal's template. Furthermore, the line spacing is not uniform.
Revision_2: I suggest the authors include the full names in the abstract (XDR, MDR and PDR), as in this form it is not immediately understandable.
Revision_3: In the title you talk about complexation, but Molecular docking is not sufficient to confirm the interaction of the drug with cyclodextrins. Further studies in solution and solid state, and in addition morphological studies, are needed to confirm the theoretical studies. it is fundamental to calculate the stability constant of the complex (Critically important request).
I suggest citing and following this recent paper, in which a careful characterization was carried out https://doi.org/10.3390/pharmaceutics15092209
Revision_4: In the inclusion complex preparation section, the molar ratio of the complexes is not reported. Why did you decide to weigh the cyclodextrins and the drug equally? (Critically important request).
Revision_5: Regarding figure 2, is this your study? Otherwise, you must provide the references.
Revision_6: Equations and tables must be in the format required by the journal.
Author Response
The authors would like to thank the reviewer for time and efforts invested to review the manuscript. We believe the quality of the manuscript has been improved accordingly. We carefully and respectfully responded to all comments raised and point-to-point responses are as below:
Dear author,
The paper entitled “Comparative investigation into The Roles of Imipenem:Cyclodextrin Complexation and Antibiotic Combination in Combatting Antimicrobial Resistance in Gram-Negative Bacteria” has been intensively reviewed and evaluated. Although present study was considered an interesting study, there were some points that need to be revised. Hereby, I would like to present my suggestions and revisions.
Revision_1: before publication, the paper must be formatted according to the journal's template. Furthermore, the line spacing is not uniform.
The manuscript formatting has been revised and line spacing was adjusted accordingly.
Revision_2: I suggest the authors include the full names in the abstract (XDR, MDR and PDR), as in this form it is not immediately understandable.
Full name was added for each in the abstract (the first cite they appeared)
Revision_3: In the title you talk about complexation, but Molecular docking is not sufficient to confirm the interaction of the drug with cyclodextrins. Further studies in solution and solid state, and in addition morphological studies, are needed to confirm the theoretical studies. it is fundamental to calculate the stability constant of the complex (Critically important request).
The Energy scores has been investigated and the energy scores indicate the stability of complexes. Further characterization technique in solid state has now been provided. New FTIR spectra are shown in Figure 2 and Figure 3. The stability constant is calculated from solubility studies for poorly soluble drugs; however imipenem is water soluble and the complexation with cyclodextrin was in purpose of enhancing stability.
I suggest citing and following this recent paper, in which a careful characterization was carried out https://doi.org/10.3390/pharmaceutics15092209
The reference has now been cited.
Revision_4: In the inclusion complex preparation section, the molar ratio of the complexes is not reported. Why did you decide to weigh the cyclodextrins and the drug equally? (Critically important request).
Molar ratio (1:1) has been studied and prepared by weighing the equivalent of moles in weight. The exact amounts of drug and cyclodextrins have now been provided.
Revision_5: Regarding figure 2, is this your study? Otherwise, you must provide the references.
Response: It represents our results and the original excel sheet was attached (Suppl. File 1)
Revision_6: Equations and tables must be in the format required by the journal.
Response: Tables and equations was performed according to the journal requirements
Reviewer 2 Report
The paper approaches a very actual and important challenge of antimicrobial therapy regarding the fight against antibiotic-resistant microorganisms. The selected Gram-negative species are of interest, belonging to frequent and dangerous infections, as well as the association of the aminoglycoside antibiotic with the cyclodextrin-carbapenem complex.
A comprehensive literature description supports the topic. The results are well supported by the appropriate application of the experimental methods especially in the microbiology section, but the computer-designed cyclodextrin-imipenem complexes require some analytical characterization.
Some editorial mistakes need to be corrected.
Author Response
The authors would like to thank the reviewer for time and efforts invested to review the manuscript. We believe the quality of the manuscript has been improved accordingly. We carefully and respectfully responded to all comments raised and point-to-point responses are as below:
The paper approaches a very actual and important challenge of antimicrobial therapy regarding the fight against antibiotic-resistant microorganisms. The selected Gram-negative species are of interest, belonging to frequent and dangerous infections, as well as the association of the aminoglycoside antibiotic with the cyclodextrin-carbapenem complex.
A comprehensive literature description supports the topic. The results are well supported by the appropriate application of the experimental methods especially in the microbiology section, but the computer-designed cyclodextrin-imipenem complexes require some analytical characterization.
Further solid-state characterization including FTIR spectroscopy has now been provided. New two figures (Figure 2 and Figure 3) were added.
Some editorial mistakes need to be corrected.
The whole manuscript has been revised.
Reviewer 3 Report
The current study aimed to compare the effect of two different combinations (imipenem and amikacin/imipenem and cyclodextrins complexes) on two pathogens which were isolated from patients with various infections in Egypt.
The study is interesting, as well as the results, and it seems that a considerable amount of work has been involved in its elaboration.
However, the quality of the presentation must be improved, as there are many paragraphs which are very difficult to understand (including in the materials and methods section), such as (but not limited to):
-Abstract: Lines 18-19; Line 26 and 27-the words respectively and performed
-Lines 57-59: the paragraph does not make sense in this form
-Lines 61: aminoglycosides are not is
-Lines 68-69: should be rephrased
-Line 107: you should define the abbreviation first, before using it in the text
-Lines 133-134: difficult to understand
-Line 150: the first sentence has no meaning
-Line 249: should be rephrased
-Line 253-255: difficult to understand
-Line 261: clavulanic acid
-Line 355: the disc diffusion method was studied??? Maybe employed, or used would be better.
I strongly recommend to revise the English language, as there are many paragraphs which are difficult to understand, which greatly impacts the presentation of this work.
Author Response
The authors would like to thank the reviewer for time and efforts invested to review the manuscript. We believe the quality of the manuscript has been improved accordingly. We carefully and respectfully responded to all comments raised and point-to-point responses are as below:
The current study aimed to compare the effect of two different combinations (imipenem and amikacin/imipenem and cyclodextrins complexes) on two pathogens which were isolated from patients with various infections in Egypt.
The study is interesting, as well as the results, and it seems that a considerable amount of work has been involved in its elaboration.
However, the quality of the presentation must be improved, as there are many paragraphs which are very difficult to understand (including in the materials and methods section), such as (but not limited to):
The whole manuscript has been revised and edited for clarity
-Abstract: Lines 18-19; Line 26 and 27-the words respectively and performed
The words have been corrected.
-Lines 57-59: the paragraph does not make sense in this form
The paragraph has now been modified accordingly.
-Lines 61: aminoglycosides are not is
The sentence has now been corrected.
-Lines 68-69: should be rephrased
The paragraph has now been revised and rephrased.
-Line 107: you should define the abbreviation first, before using it in the text
The full words of abbreviation were mentioned.
-Lines 133-134: difficult to understand
The whole paragraph has now been revised.
-Line 150: the first sentence has no meaning
The whole paragraph has now been revised.
-Line 249: should be rephrased
It has now been rephrased.
-Line 253-255: difficult to understand
The whole paragraph has now been revised.
-Line 261: clavulanic acid
The word acid has now been included.
-Line 355: the disc diffusion method was studied??? Maybe employed, or used would be better.
It has now been modified accordingly.
I strongly recommend to revise the English language, as there are many paragraphs which are difficult to understand, which greatly impacts the presentation of this work.
The whole manuscript has now been revised and rewritten.
Reviewer 4 Report
Introduction
· Line 65: There are conflicting reports on the sensitivity of XDR for Gram-negative bacteria,
What the authors mean by "there are conflicting reports" needs to be clarified.
· In the introduction the authors need to write about imipenem and amikacin combination
Materials and methods:
· Lines 335-338: need more details about the method, such as quantity of hydro-methanolic solution added, the duration before drying, ambient conditions
· Why the author used hydro-methanolic solution (50% v/v) ??
· Lines 332 to 338: Characterization and or testing of imipenem, -CD, and HP -CD physical combinations produced by the two methods would be beneficial in ensuring their stability, complex formation, and solubility.
Results:
· Are the data in Table 1 from the literature, or were these measurements performed by the authors? Please clarify and include a reference in the table if the results are from the literature.
· Figure 1. Needed reference
· Figure 2: It is advisable to organize the figures in descending order
Discussion
· The effect of CDs complexes produced with -CD and HP-CD prepared using two techniques physical mixing and kneading methods must be shown, compared, and discussed in the results and discussion.
· It is necessary to discuss the compression effect of both -CD and HP -CD.
· In the discussion, the authors included a significant number of previous studies that are general regarding the action of amikacin and imipenem, both alone and in combination, and these data are irrelevant to the core of the present study. It is important to rewrite the discussion and focus on the current study findings.
· This is on line 372,373. The OD cannot be the number of organisms. The OD is measured at 600 nm on a scale from 0 to 1 and this can be correlated with the number of organisms.
Author Response
The authors would like to thank the reviewer for time and efforts invested to review the manuscript. We believe the quality of the manuscript has been improved accordingly. We carefully and respectfully responded to all comments raised and point-to-point responses are as below:
Line 65: There are conflicting reports on the sensitivity of XDR for Gram-negative bacteria,
What the authors mean by "there are conflicting reports" needs to be clarified.
Response: the sentence was rewritten to be clarified
- In the introduction the authors need to write about imipenem and amikacin combination
Response: a paragraph was added to the introduction about imipenem and amikacin combination (Line 64)
Materials and methods:
- Lines 335-338: need more details about the method, such as quantity of hydro-methanolic solution added, the duration before drying, ambient conditions
- Why the author used hydro-methanolic solution (50% v/v) ??
Hydromethanolic solution was used to be easily evaporated at ambient conditions and this mixture system is a common solvent for both cyclodextrin and imipenem and hence allows intimate interactions (molecular interactions).
- Lines 332 to 338: Characterization and or testing of imipenem, -CD, and HP -CD physical combinations produced by the two methods would be beneficial in ensuring their stability, complex formation, and solubility.
The complex formation of imipenem with cyclodextrin was for improving antimicrobial activity. Solubility studies are not beyond the scope of this manuscript as the drug is soluble in water but suffer from stability problems; therefore complexation could enhance stability. The complex formation was confirmed by two methods in silico (molecular docking) and FTIR spectroscopy.
Results:
- Are the data in Table 1 from the literature, or were these measurements performed by the authors? Please clarify and include a reference in the table if the results are from the literature
- Figure 1. Needed reference
The results shown in Figure 1 and Table 1 are original and generated from the docking studies.
- Figure 2: It is advisable to organize the figures in descending order
Response: it is difficult to do that as results were represented for 2 microbial species that have different response for each antibiotic.
Discussion
- The effect of CDs complexes produced with -CD and HP-CD prepared using two techniques physical mixing and kneading methods must be shown, compared, and discussed in the results and discussion.
There were no statistical significance differences between Beta-CD and HP-Beta CD
- It is necessary to discuss the compression effect of both -CD and HP -CD.
We have not compressed both CDs; we used the dispersed mixtures instead.
- In the discussion, the authors included a significant number of previous studies that are general regarding the action of amikacin and imipenem, both alone and in combination, and these data are irrelevant to the core of the present study. It is important to rewrite the discussion and focus on the current study findings.
Response: it was revised and modified.
- This is on line 372,373. The OD cannot be the number of organisms. The OD is measured at 600 nm on a scale from 0 to 1 and this can be correlated with the number of organisms.
Response: the sentence was corrected.
Reviewer 5 Report
The manuscript demonstrates a comparative study between the effects of imipenem and amikacin combination and impinem with cyclodextrin complexes (β-cyclodextrin and hydroxypropyl β-cyclodextrin) on two of the most important pathogens (Acinetobacter baumannii and Klebsiella pneumoniae) isolated from patients with various infections in the upper Egypt region. The idea sounds interesting, which, however, requires revision to further improve the quality of the manuscript.
Provide a schematic explaining the outline of the study.
The article lacks characterization of the successful formation of complexes. Suggest providing more characterizations. such as FTIR.
I suggest providing the the spread plate method results of the eventual formulation.
The authors mentioned safe biofunctional excipient. It should be demonstrated using atleast viability of human cells in their presence.
Better cite recent past 3 years refs on this topic.
The manuscripts needs extensive editing. Some examples are here.
The manuscript contains irregularly placed hypens dividing words. for example "per-formed".
Abbreviations must be defined at their first appearance in the abstract and main text.
By the way, several grammar mistakes in most of the sentences.
Author Response
The authors would like to thank the reviewer for time and efforts invested to review the manuscript. We believe the quality of the manuscript has been improved accordingly. We carefully and respectfully responded to all comments raised and point-to-point responses are as below:
Minor revision The manuscript demonstrates a comparative study between the effects of imipenem and amikacin combination and impinem with cyclodextrin complexes (β-cyclodextrin and hydroxypropyl βcyclodextrin) on two of the most important pathogens (Acinetobacter baumannii and Klebsiella pneumoniae) isolated from patients with various infections in the upper Egypt region.
The idea sounds interesting, which, however, requires revision to further improve the quality of the manuscript. Provide a schematic explaining the outline of the study.
The article lacks characterization of the successful formation of complexes. Suggest providing more characterizations. such as FTIR.
The FTIR spectra for the drug, Cds and their complexes have now been provided. Two new figures 2 and 3 were included.
I suggest providing the the spread plate method results of the eventual formulation.
Response: Results of Time kill assay were attached (Suppl. File 2)
The authors mentioned safe biofunctional excipient. It should be demonstrated using at least viability of human cells in their presence. Better cite recent past 3 years refs on this topic.
Cyclodextrins are cyclic sugars and their safety has been studied extensively. Relevant references in the last three years have been provided .
The manuscripts needs extensive editing. Some examples are here.
The manuscript contains irregularly placed hypens dividing words. for example "per-formed".
Abbreviations must be defined at their first appearance in the abstract and main text.
By the way, several grammar mistakes in most of the sentences.
The whole manuscript has now been revised and rewritten.
Round 2
Reviewer 1 Report
I accept the work in this form.
Author Response
The authors are very pleased that we have addressed all the comments.
Reviewer 3 Report
I believe the recommendations/suggestions mentioned in the first review report are now fulfilled and the presentation of this word was clearly improved.
Author Response
The authors would like to thank the reviewer for the positive comments.
Reviewer 4 Report
since the scope of the current study "This study was carried out to assess combination therapy for imipenem and amikacin in treatment of XDR Gram negative pathogens in comparison with monotherapy using checkerboard, and time–kill plots in comparison to antibiotic the effect of imipenem:CDs complexes with β-CD and HP β-CD prepared using two methods physical mixing and kneading methods"
despite the fact that there were no statistically significant differences, the author must describe these results in the discussion and in the conclusion.
Author Response
since the scope of the current study "This study was carried out to assess combination therapy for imipenem and amikacin in treatment of XDR Gram negative pathogens in comparison with monotherapy using checkerboard, and time–kill plots in comparison to antibiotic the effect of imipenem:CDs complexes with β-CD and HP β-CD prepared using two methods physical mixing and kneading methods"
despite the fact that there were no statistically significant differences, the author must describe these results in the discussion and in the conclusion.
This is a valid point as it is worth mentioning to emphasize these interesting findings. The authors would like to thank the authors for that.